# COLLABORATIVE-REVERSE DIFFUSION MODELS

## ABSTRACT

Given sufficient training samples, diffusion models have outperformed in various data generation tasks. However, in the domain of natural science, researches are often challenged by class imbalance and data sparseness. In order to apply the diffusion models to general natural science scenarios, we propose the Collaborative-Reverse Diffusion Models (CoReDM), a novel diffusion-based framework that incorporates a neural collaborative mechanism to improve the generating ability for the sub-tasks with sparse data. The CoReDM features a new paradigm to integrate a collective similarity regularization into the reverse diffusion processes. This design facilitates adaptive knowledge transfer for the denoising processes among sub-tasks. Experimental results on several imbalanced datasets demonstrate that our CoReDM substantially enhances both the quality and diversity of generated samples, outperforming conventional diffusion models and a few data augmentation baselines.

## 1 INTRODUCTION

Diffusion models have attracted significant attention due to their excellent generation quality and high flexibility. Based on the diffusion models, various applications have emerged, such as high-resolution image generation Sohl-Dickstein et al. (2015); Song & Ermon (2019), text-to-audio Huang et al. (2023), and video generation Ho et al. (2022), etc. Despite the success of current diffusion models, there are still some limitations in the diffusion framework. For the sparse and small datasets, the conventional diffusion models often have difficulty to learn effective features, which affects the quality of the generated results. Particularly in scenarios with co-occurring label imbalance (e.g., rare material categories) and data sparsity (e.g., limited experimental measurements).

Some researchers have tried to alleviate the negative impact of sparse data on the performance of diffusion models by introducing various sampling strategies and optimizing the loss function. However, existing solutions mainly focus on improving the performance of the available samples ,while there is still a lack of discussion on how to effectively use information from similar categories for learning. This oversight is a notable problem in scientific domains, because data collection is expensive and time-consuming. Knowledge transfer between related categories could be a promising direction. Thus, we are motivated to consider how to enhance the sample learning process by learning features from similar categories.

This work presents the Collaborative-Reverse Diffusion Models (CoReDM), which integrates a collaborative filtering (CF) mechanism into the reverse diffusion process. To improve the generation quality of sparse classes with limited samples, our model dynamically utilizes semantic relationships between samples to guide the denoising trajectory. Specifically, a neural collaborative regularization term is introduced to evaluate inter-sample influence and adaptively adjusts collaborative weights. This design facilitates effective knowledge transfer from data-rich categories to data-scarce ones by refining the denoising path through relational signals, leading to significant improvements in the generative performance for underrepresented categories.

Our contributions are as follows:

- We propose a novel collaborative reverse process for diffusion models by incorporating a similar-relation mechanism. This enables similar samples to influence and guide each other's denoising trajectories, shifting the process from isolated computation toward collective reasoning.

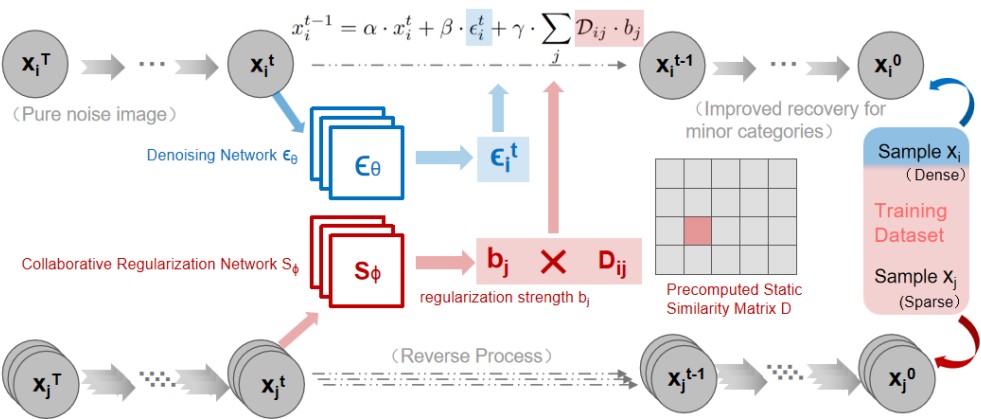

Figure 1: **Illustration of our CoReDM framework.** The CoReDM framework integrates two neural networks: a standard denoising network (blue) and a neural collaborative regularization network (red). This dual-network architecture enables knowledge transfer from dense to sparse classes during reverse diffusion, enhancing generation quality for imbalanced datasets through learned sample relationships.

- We introduce a neural collaborative regularization term that adaptively learns to assign influence weights between samples. This module provides a trainable and content-aware alternative to static relational priors, allowing the model to dynamically identify and utilize the most beneficial guidance signals during denoising.

## 2 RELATED WORK

### 2.1 DIFFUSION MODELS FOR SPARSE DATASETS.

The traditional diffusion model performs poorly on class-imbalanced datasets Yang et al. (2024); Wang et al. (2024); Xu et al. (2024). Training on such datasets leads to a notable decline in diversity and fidelity, particularly for sparse classes. In these classes, generated content loses much diversity and severe mode-collapse issues arise. To solve this problem, Yan et al. Yan et al. (2024) proposed a framework based on probabilistic contrastive learning, which minimizes distribution overlap by penalizing the KL divergence between conditional distributions of different classes. In addition to these, Qin et al. (2023) used a distribution adjustment regularizer to train the model, adjusting the conditional transition probabilities during training. However, when there is a significant difference between the empirical distribution and the label distribution, this method may affect the stability of training and lead to a decrease in performance. In order to tackle the challenge of sample scarcity from a different angle, Dhariwal & Nichol (2021) improved the quality of generation by refining model architectures and employing classifier guidance. But their method is constrained to labeled data, which is a significant limitation. Drawing inspiration from the demonstrated success of collaborative filtering in recommender systems for overcoming data sparsity through similarity-based knowledge transfer (Breese et al., 2013; He et al., 2017; Koren et al., 2009; Su & Khoshgoftaar, 2009), we propose to integrate dynamic neighborhood constraints into the diffusion process.

### 2.2 APPLICATION OF GENERATIVE MODELS TO MATERIAL GENERATION.

Recent advances in generative models have opened new possibilities for material discovery. Generative models can learn structure-property relationships from physical data. Traditional approaches, such as variational autoencoders (VAEs) and generative adversarial networks (GANs), have been applied to generate crystal structures, predict phase diagrams and optimize material compositions (Chen et al., 2021; Hong et al., 2020; Li et al., 2022; Noh et al., 2020; Butler et al., 2018; Sanchez-Lengeling & Aspuru-Guzik, 2018; Ward et al., 2016). For instance, Fuhr & Sumpter (2022) demonstrated that GANs can generate novel alloy compositions by learning from high-throughput computational databases, and highlighted the potential of deep generative models in accelerating materials

discovery and innovation . However, these models often suffer from mode collapse, leading to limited diversity in generated samples (Cao et al., 2024; Goodfellow et al., 2020).

Departing from conventional approaches mentioned above, diffusion models have emerged as a powerful tool in physical materials field (Park et al., 2024; Vecchio et al., 2024; Xie et al., 2021; Alverson et al., 2024). Bastek et al. (2024) present a framework that integrates diffusion models with physical constraints, enabling the generation of materials that not only fit the desired data distribution, but also satisfy the governing physical equations. The framework addresses the limitation of traditional generative models, making it easy to directly impose constraints. This suggests that the application of diffusion models in material generation is expected to be further extended, with potential application prospects in predicting material properties.

## 3 METHOD

### 3.1 BASIC DIFFUSION MODELS

Diffusion models learn data generation through a forward noise-adding and a reverse denoising process. The forward process gradually corrupts the initial data point $x_0$ into a pure noise $x_T$ over $T$ steps. Given an initial data point $x_0 \sim q(x_0)$ and a variance schedule $\beta_t \in (0, 1)$ that controls the amount of noise added at each step, the forward process is defined as:

$$q(x_t|x_{t-1}) = \mathcal{N}(x_t; \sqrt{1 - \beta_t}x_{t-1}, \beta_t I).$$

Over the course of $T$ steps, the data $x_0$ is gradually noised into pure noise $x_T$.

The reverse process aims to recover the original data by learning to iteratively denoise. Following the DDPM framework Ho et al. (2020), the reverse process is formally defined as learning a conditional distribution:

$$p_\theta(x_{t-1} \mid x_t) = \mathcal{N}(x_{t-1}; \mu_\theta(x_t, t), \Sigma_\theta(x_t, t)),$$

where the mean $\mu_\theta(x_t, t)$ and variance $\Sigma_\theta(x_t, t)$ are parameterized by a neural network. A common parameterization is to fix the variance and use the network predict the added noise $\epsilon_\theta(x_t, t)$ in the forward process. This leads to a simplified and widely used update rule for sampling:

$$x_{t-1} = \frac{1}{\sqrt{\alpha_t}} \left( x_t - \frac{\beta_t}{\sqrt{1 - \bar{\alpha}_t}} \epsilon_\theta(x_t, t) \right) + \sigma_t z$$

where $z \sim \mathcal{N}(0, I)$ and $\sigma_t$ is the standard deviation derived from the fixed variance schedule. The training objective is therefore to minimize the error between the predicted noise $\epsilon_\theta$ and the true noise $\epsilon$ used in the forward process:

$$\mathcal{L} = \mathbb{E}_{t,x_0,\epsilon} \left[ \|\epsilon - \epsilon_\theta(x_t, t)\|^2 \right]$$

Through this learning process, the model can start from random noise $x_T$ and iteratively generate samples that resemble the original data distribution.

### 3.2 NEURAL COLLABORATIVE DENOISING DIFFUSION

In order to integrate the CF mechanism into the generation process, we propose a dual-network architecture. This architecture extends the standard diffusion model by incorporating influences from similar samples during denoising. The whole CoReDM system consists of two coordinated components: a denoising network that preserves the essential noise prediction function, and a neural collaborative regularization term that learns to measure inter-sample relevance. The output weights from the regularization term are combined with a precomputed similarity matrix to regulate the degree of guidance provided by each neighboring sample.

This design allows the model to effectively utilize collective information from related samples throughout the reverse diffusion process. The following sections describe the architecture of the neural collaborative regularization term and provide the complete formulation of the collaborative reverse process.

### 3.2.1 NEURAL COLLABORATIVE REGULARIZATION TERM DESIGN

The neural collaborative regularization term represents a critical innovation designed to address the inherent limitations of conventional diffusion models in handling samples from the sparse class. Traditional approaches process each sample in isolation during denoising, which proves particularly inadequate for underrepresented categories where insufficient data leads to degraded generation quality. Our framework introduces a learned scoring mechanism that enables knowledge transfer across semantically related samples, allowing dense-class samples to guide the denoising process of sparse-class categories through relational inference.

The core of this design is a parametric mapping function that learns static influence relationships between samples. The fundamental mathematical expression governing this mapping is defined as:

$$b_j = S_\phi(\mathbf{x}_i, \mathbf{x}_j),\tag{1}$$

where $\mathbf{x}_i$ and $\mathbf{x}_j$ denote the latent representations of samples $i$ and $j$ respectively, and $S_\phi$ represents a parameterized neural network that outputs a scalar regularization strength $b_j$. The network architecture may employ multilayer perceptrons or convolutional networks to process sample pair features.

The network parameter $\phi$ will be optimized using loss signals aggregated across all timesteps, thus the $b_j$ leverages the cumulative information from the entire diffusion spectrum rather than adapting to individual timesteps. This design encourages the model to learn stable and generalizable relationships between samples, preventing overfitting to features associated with a specific noise level.

### 3.2.2 COLLABORATIVE-REVERSE PROCESS

Building upon the scores obtained from the neural collaborative regularization term, we reformulate the reverse process of diffusion models. The proposed collaborative reverse process can be intuitively understood as a form of similarity-based guidance, where each sample's denoising trajectory is dynamically adjusted by its neighbors within the data manifold. While conventional reverse processes denoise each sample independently, our approach introduces a collaborative mechanism that allows samples to mutually guide each other's denoising trajectories.

This mechanism can be formally related to imposing a prior that minimizes the divergence between the denoising trajectories of neighboring points in the data manifold. The core update rule of the collaborative reverse process is defined as:

$$x_i^{t-1} = \alpha \cdot x_i^t + \beta \cdot \epsilon_i^t + \gamma \cdot \sum_j \mathcal{D}_{ij} \cdot b_j \cdot ((x_i^t, t) - \epsilon_\theta(x_j^t, t)),\tag{2}$$

where $\alpha \cdot x_i^t + \beta \cdot \epsilon_i^t$ constitutes the standard denoising term, $\mathcal{D}_{ij}$ denotes the precomputed static similarity matrix, constructed from feature embeddings of the training samples using cosine similarity, which provides a fixed relational prior. $b_j$ is the influence score generated by the neural collaborative regularization term, and $\gamma$ is a hyperparameter controlling the collaboration strength. The summation over $j$ in the collaborative term leads to a computational complexity of $O(n^2)$ for the reverse process, where $n$ is the batch size.

The collaborative term in this formulation enables semantic relationship-based denoising guidance. For each target sample $i$, its denoising direction is determined not only by its own state but also by the weighted influence of all other samples $j$. The weights are jointly determined by static similarity $\mathcal{D}_{ij}$ and dynamically learned influence $b_j$, ensuring that semantically similar samples exert a more significant influence on the denoising processes of the other.

### 3.3 TRAINING STRATEGY

To jointly optimize the denoising network and the collaborative regularization network, we employ a two-phase alternating training strategy that enhances stability by alternately fixing one network's parameters while training the other.

The composite loss function combines standard diffusion loss with collaborative consistency loss:

$$\mathcal{L}_{\text{total}} = \mathcal{L}_{\text{diffusion}} + \lambda(t) \cdot \mathcal{L}_{CF}, \quad \lambda(t) = \lambda_{\max} \cdot (1 - \frac{t}{T})\tag{3}$$

---

**Algorithm 1** Collaborative-Reverse Diffusion Models Training

---

**Require:** Training dataset $\mathcal{D}$, Noise schedule $\{\beta_t\}_{t=1}^T$, Similarity matrix $\mathcal{D}_{ij}$, Loss balancing coefficient $\lambda$

**Ensure:** Optimized parameters $\theta$ (denoising network) and $\phi$ (collaborative regularization network)

1: Initialize $\theta$ and $\phi$
2: **for** iteration = 1 to MaxIterations **do**
3:    **— Phase 1: Denoising Network Update —**
4:    Freeze $\phi$, sample $\mathbf{x}_0 \sim \mathcal{D}$, $t \sim \text{Uniform}(1, T)$, $\epsilon \sim \mathcal{N}(0, \mathbf{I})$
5:    $\mathbf{x}_t \leftarrow \sqrt{\bar{\alpha}_t}\mathbf{x}_0 + \sqrt{1 - \bar{\alpha}_t}\epsilon$ {Noisy sample}
6:    $\hat{\epsilon} \leftarrow \epsilon_\theta(\mathbf{x}_t, t)$ {Noise prediction}
7:    $\mathcal{L}_{\text{diffusion}} \leftarrow \|\epsilon - \hat{\epsilon}\|^2$ {Prediction loss}
8:    $\mathcal{L}_{\text{CF}} \leftarrow \sum_{i,j} \mathcal{D}_{ij} \cdot b_j \cdot \left\|\epsilon_\theta(x_i^t, t) - \epsilon_\theta(x_j^t, t)\right\|^2$
      {Compute consistency loss for sampled timestep t}
9:    Update $\theta \leftarrow \theta - \eta_\theta \nabla_\theta(\mathcal{L}_{\text{diffusion}} + \lambda \cdot \mathcal{L}_{\text{CF}})$
10:   **— Phase 2: Collaborative Regularization Network Update —**
11:    Freeze $\theta$, sample new $\mathbf{x}_0 \sim \mathcal{D}$, $t \sim \text{Uniform}(1, T)$
12:    $b_j \leftarrow S_\phi(\mathbf{x}_i, \mathbf{x}_j)$ {Collaborative regularization}
13:    $\mathcal{L}_{\text{CF}} \leftarrow \sum_{i,j} \mathcal{D}_{ij} \cdot b_j \cdot \left\|\epsilon_\theta(x_i^t, t) - \epsilon_\theta(x_j^t, t)\right\|^2$
      {Compute consistency loss for sampled timestep t}
14:   Update $\phi \leftarrow \phi - \eta_\phi \nabla_\phi \mathcal{L}_{\text{CF}}$
15: **end for**
16: **return** optimized parameters $\theta^*, \phi^*$

---

, where $\lambda_{\text{max}}$ is the maximum weight, $T$ is the total diffusion steps, and $\mathcal{L}_{\text{diffusion}} = \mathbb{E}_{t,\mathbf{x}_0,\epsilon}\left[\|\epsilon - \epsilon_\theta(\mathbf{x}_t, t)\|^2\right]$ ensures accurate noise prediction. The $\mathcal{L}_{\text{CF}}$ is defined as:

$$\mathcal{L}_{CF} = \mathbb{E}_t\left[\sum_{i,j} \mathcal{D}_{ij} \cdot b_j \cdot \left\|\epsilon_\theta(x_i^t, t) - \epsilon_\theta(x_j^t, t)\right\|^2\right] \tag{4}$$

The time-dependent weighting $\lambda(t) = \lambda_{\text{max}} \cdot (1 - t/T)$ progressively strengthens CF constraints, as t decreases and the noisy input becomes cleaner. Because early steps with high noise levels require more freedom for global structure recovery, while later steps with clearer images benefit from stronger similarity constraints for local refinement.

The training procedure begins by freezing the collaborative regularization network parameter $\phi$, optimizing only the denoising network parameters $\theta$. This phase ensures that the denoising network learns to accurately predict noise within the established collaborative framework. Specifically, we randomly sample timestep $t$, original data $\mathbf{x}_0$, and noise $\epsilon$, compute $\mathbf{x}_t$, and feed it into the networks to obtain predicted noise $\epsilon_\theta(\mathbf{x}_t, t)$. The gradient of the total loss with respect to $\theta$ is then computed:

$$\nabla_\theta \mathcal{L}_{\text{total}} = \nabla_\theta \mathcal{L}_{\text{diffusion}} + \lambda \cdot \nabla_\theta \mathcal{L}_{\text{CF}} \tag{5}$$

The denoising network parameters are updated using stochastic gradient descent with learning rate $\eta_\theta$ (e.g., Adam optimizer):

$$\theta \leftarrow \theta - \eta_\theta \cdot \nabla_\theta \mathcal{L}_{\text{total}}$$

In the second phase, we fix the denoising network parameters $\theta$ and optimize only the collaborative regularization network parameters $\phi$. Only the gradient of the collaborative consistency loss with respect to $\phi$ is computed:

$$\nabla_\phi \mathcal{L}_{\text{CF}} = \nabla_\phi\left[\sum_{i,j} \mathcal{D}_{ij} \cdot b_j \cdot \left\|\epsilon_\theta(x_i^t, t) - \epsilon_\theta(x_j^t, t)\right\|^2\right] \tag{6}$$

The collaborative regularization network parameters are updated similarly:

$$\phi \leftarrow \phi - \eta_\phi \cdot \nabla_\phi \mathcal{L}_{\text{CF}}$$

This phase minimizes the collaborative consistency loss, training the collaborative regularization network to identify sample pairs with consistent noise predictions and dynamically adjust their influence weights $b_j^t$, enhancing collaborative guidance among similar samples.

The two phases are executed alternately, achieving collaborative evolution of both networks through iterative optimization, and ultimately balancing generation quality with inter-sample consistency. Learning rates $\eta_\theta$ and $\eta_\phi$ can be dynamically adjusted during training, with $\eta_\theta$ typically set larger than $\eta_\phi$ to ensure prioritized convergence of the denoising network. Through iterative alternation between these two phases, both networks co-evolve to achieve the dual objectives of maintaining sample generation quality while enhancing inter-sample consistency. This training strategy effectively addresses the stability challenges associated with joint network optimization while ensuring effective learning of the collaborative mechanism.

## 4 EXPERIMENT

### 4.1 EXPERIMENTAL SETUP

#### 4.1.1 DATASET AND PREPROCESSING

Our experimental framework employs a comprehensive material property dataset characterized by a significant class imbalance, where certain material categories (e.g., high-entropy alloys and complex oxides) are substantially underrepresented compared to conventional metallic compounds. This dataset incorporates two primary feature categories: Input Features, serving as conditional information, include compositional descriptors (comprising 65 elemental, represented as fractions) and density. Output Features, serving as the generation target, consist of thermal properties (thermal conductivity and thermal capacity), which are the target properties the model learns to reconstruct from noise. A more detailed description of the imbalance and sparsity of this dataset can be found in the appendix. During the experiment process, the dataset is divided into training (70%) and validation (30%) subsets. The partitioning methodology enforces strict preservation of original class proportions throughout the dataset division process, which carefully designed to prevent arbitrary splits that could distort results.

To utilize the intuitive similarity relationship, we precompute a static similarity matrix $D_{ij}$, which embeds relational priors. We first obtain a feature representation $\mathbf{x}_i'$ for each raw sample $\mathbf{x}_i$ using a pre-trained encoder, where $\mathbf{x}_i' = \mathbf{Encoder}(\mathbf{x}_i)$. The pairwise similarity is then computed as the cosine similarity between these representations:

$$\mathcal{D}_{ij} = \frac{\mathbf{x}_i' \cdot \mathbf{x}_j'}{\|\mathbf{x}_i'\|\|\mathbf{x}_j'\|} \tag{7}$$

The $D_{ij}$ remains fixed throughout training, providing a stable foundation for the subsequent dynamic collaboration guided by the neural regularization term.

The model architecture builds upon dual-network, which consists of a denoising U-Net and a neural collaborative regularization term. The denoising network adopts the standard DDPM structure, while the collaborative regularization term is implemented as a three-layer MLP with 256 hidden units per layer and ReLU activation. The model is trained using alternating updates: the denoising network is optimized with AdamW (learning rate = $2e^{-4}$), and the collaborative regularization term is trained with Adam (learning rate = $e^{-4}$). Gradient clipping (max norm = 1.0) is applied to the regularization term to ensure training stability.

#### 4.1.2 BASELINE METHOD

To evaluate the proposed CoReDM framework, we design several baseline methods that progressively dissect the contribution of its core collaborative components. In addition to the Class-Balancing Diffusion Model (CBDM) Qin et al. (2023) as a representative baseline for imbalanced datasets, we first implement a standard diffusion model (No-Collab) that sets the $\gamma$ to zero. This baseline preserves all other architectural elements while generating each sample independently through conventional reverse diffusion:

$$x_i^{t-1} = \alpha \cdot x_i^t + \beta \cdot \epsilon_i^t \tag{8}$$

Table 1: Performance Comparison of Collaborative Filtering Strategies

| Model | MSE↓ | $R^2$↑ | Pearson R↑ | $F_\beta$↑ | Recall↑ |
|---|---|---|---|---|---|
| No_Collab | 14.662 | 0.3125 | 0.3080 | 0.5630 | 0.5570 |
| CBDM | 8.8042 | 0.3913 | 0.5058 | 0.5797 | 0.5966 |
| Static_Collab | 10.8591 | 0.5471 | 0.5906 | 0.6248 | 0.5738 |
| Linear_Collab | **8.6399** (-6.0221) | **0.6832** (+0.3707) | **0.7050** (+0.397) | **0.6167** (+0.0537) | **0.6014** (+0.0444) |

**Note:** The values in the brackets represent relative gains.

We then introduce Static-Collab, which incorporates fixed collaboration through precomputed feature similarities $D_{ij}$ and neural weighting term $b_j$, but employs a constant collaboration strength $\gamma$ that does not vary over time:

$$x_i^{t-1} = \alpha \cdot x_i^t + \beta \cdot \epsilon_i^t + \gamma \cdot \sum_j \mathcal{D}_{ij} \cdot b_j \cdot ((x_i^t, t) - \epsilon_\theta(x_j^t, t)) \qquad (9)$$

Finally, the Linear-Collab implements a time-increasing collaboration strategy where the influence of neighboring samples grows linearly with the denoising progress by scaling the collaboration strength $\gamma$ with a time-dependent factor:

$$x_i^{t-1} = \alpha \cdot x_i^t + \beta \cdot \epsilon_i^t + \gamma \cdot (1 - t/T) \cdot \sum_j \mathcal{D}_{ij} \cdot b_j \cdot ((x_i^t, t) - \epsilon_\theta(x_j^t, t)) \qquad (10)$$

where $T$ denotes the total number of diffusion timesteps. This progression from no collaboration to constant-strength and then time-increasing collaboration allows us to methodically evaluate the impact of different collaborative integration strategies.

To ensure fair comparisons, all models employ identical network architectures and hyperparameters, including: 1,000 diffusion steps with a linear $\beta$ noise schedule, an Adam optimizer ($\eta = 10^{-4}$) with cosine annealing learning rate scheduling, and consistent training protocols (batch size=32, 100 epochs). By systematically controlling experimental variables, this approach enhances the interpretability of observed performance differences, with collaborative integration strategies identified as the primary contributing factor.

### 4.1.3 EVALUATION PROTOCOL

To rigorously assess model performance while ensuring methodological consistency, we established a comprehensive evaluation framework. All experiments are conducted with fixed random seeds (0-4) across all five independent training runs to ensure exact reproducibility.

Our evaluation employs five complementary metrics to comprehensively assess model performance: (1) Mean Squared Error (MSE) (Gupta et al., 2009); (2) $F_\beta$-score ($\beta$=1) (Sajjadi et al., 2018); (3) Recall (Kynkäänniemi et al., 2019); (4) $R^2$ evaluates the proportion of variance explained by the model; and (5) Pearson correlation coefficient ($R$) (Saccenti et al., 2020). This multi-faceted approach enables simultaneous assessment of prediction accuracy, classification robustness, and statistical correlation.

To account for stochastic variations in neural network training and initialization, we conduct five independent training runs for each experimental configuration. Performance metrics are aggregated across these runs and reported as mean ± standard deviation values.

### 4.2 MAIN RESULTS

In this part, our experimental results demonstrate three significant findings on the CoReDM. First, all collaborative variants substantially outperform the standard diffusion baseline, and the Linear_Collab model surpasses all the indicators of CBDM. As shown in Table 1, the Linear_Collab model reduces MSE (8.6399,-6.0221) and improves R² (0.6832,+0.3707), reflecting a major enhancement in predictive precision. It also attains the highest Pearson correlation (0.7050, +0.397 gain) and recall (0.6014, +0.0444), confirming its effectiveness in capturing underlying data relationships and representing scarce classes. While the Static_Collab variant also shows consistent

improvements over both the No_Collab baseline and the CBDM model across all metrics, its gains remain more moderate, confirming that the time-aware design of Linear_Collab yields superior adaptation and knowledge transfer throughout the denoising process.

Second, the performance progression from No_Collab to Static_Collab and finally Linear_Collab corroborates our design principle that adaptive, time-aware collaboration is essential for effective reverse diffusion. As evidenced in Figure 2, the Linear_Collab variant (green) demonstrates superior optimization stability, achieving lower final loss compared to both the Static_Collab (orange) and the baseline (blue).

Third, the CoReDM model demonstrates superior performance gains over conventional data augmentation methods, without compromising computational efficiency. As shown in Figure 3, we assess generation performance using a weighted score, where higher composite scores indicate better overall performance. To ensure a fair cross-model comparison, our composite weighted score combines 70% $R^2$ for goodness-of-fit with 30% normalized MSE for prediction error, in which the normalization of MSE follows 1 - MSE/max_MSE.

Compared with VAE and GAN, the CoReDM model achieves an optimal balance between physical constraint compliance and computational tractability. Our optimized collaborative regularization implementation enables both faster training and higher-quality sample generation, which outperforms alternative augmentation approaches. The collaborative term introduces a manageable overhead, increasing training time by approximately 15-20% and memory usage by less than 10% due to the efficient MLP architecture of $S_\phi$ and the sparse k-NN graph used for $\mathcal{D}_{ij}$. This demonstrates the practical feasibility of our approach.

These experimental results collectively demonstrate that integrating neural collaborative regularization into diffusion models (CoReDM) effectively addresses key challenges in data-scarce scenarios. The time-dependent collaborative strategy maintains stable gradient updates during denoising while optimally balancing computational efficiency and physical constraint compliance.

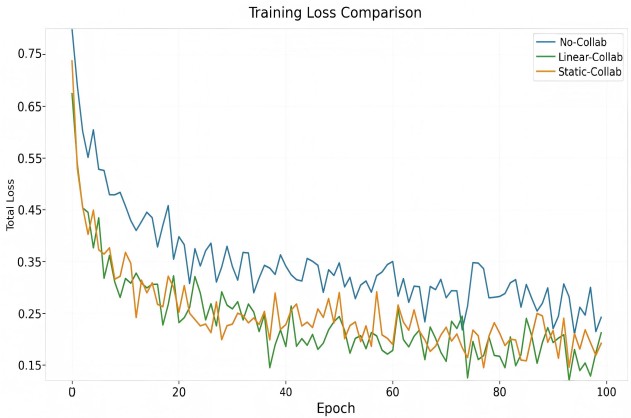

Figure 2: Training loss trajectories for baseline and CoReDM models on our material generation dataset. The proposed Linear-Collab variant (green) achieves lower final loss than both Static-Collab (orange) and baseline No-Collab (blue) approaches, demonstrating its optimization.

### 4.3 QUALITATIVE ANALYSIS

#### 4.3.1 CASE STUDY ON REPRESENTATIVE MATERIAL

For multicomponent alloys where composition and density serve as inputs, as demonstrated in Table 3 in the Appendix, the neural collaborative regularization mechanism in CoReDM consistently generates more physically consistent property predictions compared to the baseline diffusion model. The baseline predictions exhibit characteristic deviations, including extreme outliers and non-compliant predictions. In contrast, CoReDM demonstrated a better generation quality that was closer to the actual physical data. Its predictions (e.g., 69.86 versus the true value of 69.48 for Cu2Se thermal conductivity) demonstrate close alignment with true data, while maintaining monotonic trends (e.g., predictions in the 0.51–0.54 range for target values near 0.57). This improvement is particularly evident in the simultaneous prediction of electrical conductivity, thermal expansion coefficient, and yield strength.

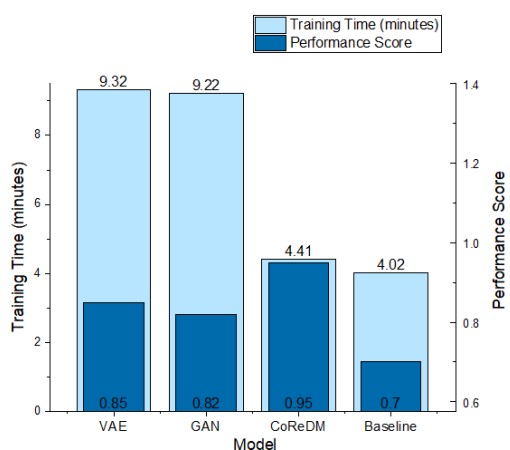
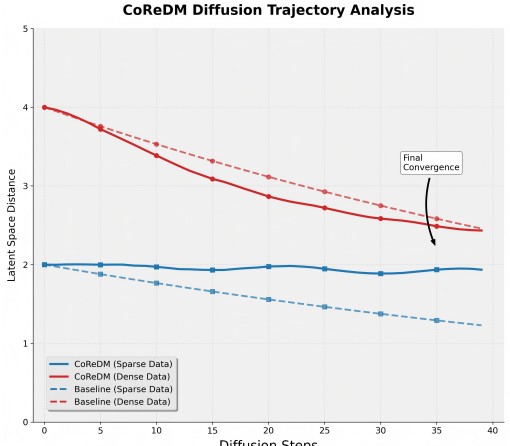

Figure 3: Comparative analysis of model efficiency and performance across different data augmentation approaches. Dark bars represent training time (left axis), while light bars indicate composite performance scores (right axis, combining R² and normalized MSE metrics). The proposed CoReDM achieves a favorable balance between computational efficiency and generative quality, outperforming other data augmentation methods.

Figure 4: Diffusion trajectory comparison between CoReDM and baseline models. Solid and dashed lines show CoReDM and baseline trajectories respectively, with blue representing sparse data and red representing dense data. The shaded region demonstrates CoReDM's guidance phase, where knowledge transfer occurs from dense to sparse data through collaborative filtering.

### 4.3.2 TRAJECTORY ANALYSIS OF KNOWLEDGE TRANSFER

To investigate the knowledge transfer mechanism of CoReDM, we visualize and analyze the diffusion trajectories of both sparse and dense classes during the denoising process, as shown in Figure 4. During the denoising phase, sparse class samples exhibit a significant upward trajectory, converging toward the dense class distribution. This convergence suggests effective knowledge transfer from abundant to rare classes through the collaborative filtering mechanism. This trajectory analysis provides direct evidence that CoReDM can effectively leverage abundant class knowledge to guide the generation of rare class samples.

### 4.3.3 TEXT TO IMAGE GENERATION VERIFICATION

To validate CoReDM's collaborative generation capabilities, we adapt its core methodology to the image domain. The experimental setup maintains an identical architecture (including UNet backbone, CLIP text encoder, and VAE image codec) and training data (subset of LAION-400M) as Stable Diffusion Rombach et al. (2022), ensuring a fair and controlled comparison. The only modification involves implementing CoReDM's collaborative reverse diffusion mechanism by introducing a neural collaborative regularization term into the sampling process, which preserves the essential dynamic and sample-wise adaptive characteristics of our framework.

Table 2: Model Performance Comparison

| Metric | Baseline Model | CoReDM |
|---|---|---|
| FID↓ | 16.26 | 12.17 (-33.6%) |
| Inception Score↑ | 31.24 | 34.07 (+9.1%) |
| CLIP Score↑ | 31.13 | 32.53 (+4.5%) |

**Note:** Arrows indicate the desired direction of improvement. All improvements are statistically significant (p $< 0.01$).

Quantitative evaluation demonstrates notable performance improvements (Table 2), with the augmented model achieving a 33.6% reduction in FID (from 16.26 to 12.17), a 9.1% improvement in Inception Score (from 31.24 to 34.07), and a 4.5% enhancement in CLIP Score (from 31.13 to 32.53). These results collectively confirm the method's effectiveness in enhancing both structural fidelity and semantic alignment, highlighting its practical value for applications that require high

precision and contextual consistency, such as scientific image synthesis. The cross-domain validation further supports the generalizability of the proposed approach.

Two key observations emerge from the empirical studies:

- The collaborative-reverse diffusion mechanism consistently enhances generation quality across modalities, with particularly pronounced gains in minority or sparse data regimes.
- The CoReDM framework proves effective in both continuous (e.g., material properties) and discrete (e.g., image pixels) output spaces, demonstrating its flexibility and robustness.

## 5 CONCLUSION

We present CoReDM, a novel framework that integrates collaborative reverse diffusion through neural relationship constraints. The key innovation lies in a dynamic neural regularization term that learns sample-wise influence weights, enabling adaptive knowledge transfer from dense to sparse classes during denoising. Experiments demonstrate CoReDM's effectiveness in addressing data imbalance for scientific generation tasks, particularly in material discovery and cross-domain image synthesis.

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

# A APPENDIX

Table 3: Comparison of predicted and true values between CIDM and baseline approaches

| Input Materials | Predicted Thermal-conductivity and Heat-capacity | | |
| --- | --- | --- | --- |
| | CoReDM | Baseline | True_Value |
| Cu2Se | 69.86 and 0.51 | 56.81 and 0.51 | 69.48 and 0.57 |
| High Alumina Brick | 68.73 and 0.54 | 72.58 and 0.52 | 69.48 and 0.57 |
| AISI 1019 Steel | 53.75 and 0.46 | 32.76 and 0.38 | 51.90 and 0.47 |
| AISI 1035 Steel | 54.96 and 0.47 | 32.90 and 0.42 | 51.90 and 0.49 |
| Stainless Steel | 65.80 and 0.42 | 40.52 and 0.40 | 69.48 and 0.42 |
| AISI 1022 Steel | 53.83 and 0.47 | 32.80 and 0.49 | 49.80 and 0.47 |
| Alumina Ceramic Armor Material | 31.66 and 0.59 | 33.23 and 0.67 | 27.50 and 0.57 |
| CeramTec Rubalit® 708 Alumina | 28.20 and 0.58 | 35.53 and 0.67 | 24.00 and 0.57 |
| Titanium Nitride | 64.59 and 0.57 | 70.44 and 0.58 | 69.48 and 0.57 |
| AISI 1045 Steel | 55.70 and 0.49 | 33.06 and 0.47 | 49.80 and 0.49 |
| AISI 1015 Steel | 58.75 and 0.49 | 32.36 and 0.41 | 51.90 and 0.49 |
| Butyl Rubber | 62.43 and 0.57 | 54.54 and 0.50 | 69.48 and 0.57 |
| Kevlar | 62.08 and 0.57 | 54.54 and 0.53 | 69.48 and 0.57 |
| Soap | 61.67 and 0.56 | 54.54 and 0.57 | 69.48 and 0.57 |
| 945 Steel | 60.92 and 0.56 | 54.54 and 0.57 | 69.48 and 0.57 |
| Titanium Carbide | 60.58 and 0.56 | 54.54 and 0.57 | 69.48 and 0.57 |
| Weldox 460 E Steel | 59.32 and 0.56 | 54.54 and 0.57 | 69.48 and 0.57 |
| Titanium Alloy | 74.01 and 0.39 | 27.23 and 0.41 | 63.20 and 0.21 |
| Gunning Castable | 56.70 and 0.61 | 51.72 and 0.60 | 69.48 and 0.57 |
| Steel Cable | 56.49 and 0.66 | 48.81 and 0.67 | 69.48 and 0.57 |

1) The table shows the predicted thermal conductivity (in W/m·K) and heat capacity (in J/g·K) values from CoReDM and baseline models, compared with experimental true values.

2) Composition column shows the main chemical components of each material, with steel grades showing their carbon content range.

3) Density values are given in g/cm$^3$ at room temperature (25°C).

4) For polymeric materials (Butyl Rubber, Kevlar), the repeating unit structure is shown.

5) The true values represent experimentally measured data from literature.

---

**Algorithm 2** Inference Procedure of CoReDM

---

**Require:** Trained denoising network parameters $\theta^*$, Trained collaborative regularization network parameters $\phi^*$, Precomputed similarity matrix $\mathcal{D}_{ij}$, Collaboration strength coefficient $\gamma$

**Ensure:** $\mathbf{x}_0$: Generated sample

1: Initialize $\mathbf{x}^T \sim \mathcal{N}(0, \mathbf{I})$ {Start from pure noise}
2: Load fixed parameters $\theta^*$ and $\phi^*$
3: **Initialize** $t \leftarrow T$
4: **For** $t = T$ **down to** $1$ **do:** {Reverse process}
5: Predict noise: $\epsilon_\theta \leftarrow \epsilon_{\theta^*}(\mathbf{x}^t, t)$
6: Compute collaborative regularization strength for all $j \in [1, N]$: $\quad b_j \leftarrow S_{\phi^*}(\mathbf{x}_i, \mathbf{x}_j)$
7: Update sample: $\quad x_i^{t-1} = \alpha \cdot x_i^t + \beta \cdot \epsilon_i^t + \gamma \cdot \sum_j \mathcal{D}_{ij} \cdot b_j((x_i^t, t) - \epsilon_\theta(x_j^t, t))$
8: **End for**
9: **Return** $\mathbf{x}_0$ {Final generated sample}

---

Our dataset comprises 1,471 samples distributed across 7 distinct material categories, including high-entropy alloys, complex oxides, and conventional metallic compounds. The class distribution demonstrates substantial imbalance, with the largest category containing 756 samples while the smallest category has only 7 samples, resulting in an imbalance ratio of 108:1. This severe imbalance presents significant challenges for conventional generative models, particularly in capturing the characteristics of underrepresented material types. Regarding data sparseness, the dataset exhibits considerable feature missingness, with approximately 45% of feature values being unavailable across the collection. On average, each sample contains 83.9 missing feature entries out of the total feature dimensions. This sparsity pattern reflects the practical challenges in material science data collection, where comprehensive experimental measurements are often difficult to obtain for all material compositions.

