# OpenReview forum: "Collaborative-Reverse Diffusion Models"
_ICLR.cc/2026/Conference — Submitted to ICLR 2026_

### Official Review · Reviewer_YFBt · 2025-10-29

**Soundness:** 1
**Presentation:** 1
**Contribution:** 2
**Rating:** 0
**Confidence:** 5

**Summary:**

This paper proposes Collaborative-Reverse Diffusion Models (CoReDM), which aim to improve the performance of diffusion models under class-imbalanced and data-sparse conditions. The authors introduce a neural collaborative regularization term that enables information sharing between semantically similar samples during the reverse diffusion process. The paper claims that this collaborative mechanism facilitates adaptive knowledge transfer from data-rich to data-scarce categories, leading to better generation quality and diversity. Experiments are conducted on a material property dataset and a text-to-image setting.

**Strengths:**

The paper addresses the class-imbalance problem in diffusion models, which is an important and timely topic, especially for scientific domains with limited data availability.

**Weaknesses:**

**1. Serious issues in correctness and clarity**
   - Eq. (2): b_j and D_{ij} are scalars, whereas the other terms are vectors, making the equation mathematically inconsistent.
   - Eq. (3): The definition of lambda(t) lacks an expectation over t, and its formulation contradicts the text description. L242-245 state that the collaborative constraint should become stronger as the image becomes cleaner, but the given formula implies the opposite.
   - Eq. (5): w_{ij} is undefined, and the loss does not depend on the parameter phi.
   - Alg. 1: x_i and x_j are never defined.
   - Similarity matrix D_{ij}: The derivation is not explained. It is also unclear how similar samples x_j are collected during training or inference.

**2. Insufficient experimental validation**
  - Important baselines such as CBDM [1] or other class-imbalanced diffusion models are missing.
   - The experimental datasets are limited. Common benchmarks for class imbalance (e.g., **CIFAR10-LT**) are not tested, making it hard to evaluate general applicability.
   - No visualizations on the T2I results. This prevents qualitative assessment of the claimed improvement.
   - There is no description of hyperparameter settings or training details (e.g., gamma, lambda_max, optimizer setup), which makes the experiments non-reproducible.
.

----
**Reference**

[1] Class-Balancing Diffusion Models, CVPR 2023

**Questions:**

See weaknesses.

---

> ### Author Response · Authors · 2025-11-23
>
> Dear Reviewer YFBt:
>
> First and foremost, we would like to extend our sincere gratitude for the time and effort you have dedicated to reviewing our manuscript. Herein, we provide detailed responses to each of your comments.
>
> **W1**:
> 1. The intended formulation is that the collaborative term applies a vector adjustment. The correct equation should sum over the vectorized contributions from neighbors j. The scalar weight D_{ij} * b_jscales the vector differencebetween the state of sample jand sample i(or its noise prediction). We have corrected the Eq(2):x_{i}^{t-1}=\alpha\cdot x_{i}^{t}+\beta\cdot\epsilon_{i}^{t}+\gamma\cdot\sum_{j}\mathcal{D}_{ij}\cdot b_j\cdot((x_{i}^{t},t)-\epsilon_{\theta}(x_{j}^{t},t)).
> 2. Eq. (3) has been corrected in the paper: \mathcal{L}_{\text{total}} = \mathcal{L}_{\text{diffusion}} + \lambda(t) \cdot \mathcal{L}_{CF}, \quad \lambda(t) = \lambda_{\text{max}} \cdot (1-\frac{t}{T})
> 3. We have revised the formula.\mathcal{L}_{CF}=\mathbb{E}_{t}\left[\sum_{i,j}\mathcal{D}_{ij}\cdot b_j \cdot\left\|\epsilon_{\theta}(x_{i}^{t},t)-\epsilon_{\theta}(x_{j}^{t},t)\right\|^{2}\right]
> 4. After presenting Eq(1) in Section 3.2.1, we immediately defined x_i and x_j.
> 5. We believe that the description of D_ij is sufficient, and we have also provided some additional details in the text.  The derivation and usage are as follows: 1. Derivation: The similarity matrix D_{ij} is precomputed and fixed​ before the main training starts. It is not derived from the diffusion model itself. As detailed in Section 4.1.1, we use a pre-trained model (e.g., on a large material database) to extract a feature vector for every sample in the training set. D_{ij} is then computed as the cosine similarity between the feature vectors of every pair of training samples (i, j). This provides a static, semantic similarity prior. 2. Usage during Training/Inference: During training, when we sample a batch of data, we simply lookup the precomputed D_{ij} values for all pairs of samples (i, j)within the batch (or potentially from a cached global matrix). During inference (sampling), to generate a new sample, we need to find its similar neighbors j from the training set. This is done by comparing the conditional input(e.g., a material composition) to the training set conditions and selecting the neighbors based on the same feature space used to build D_{ij}.
>
> **W2**:
>
> 1. We have included CBDM as the baseline in the main experiment, and the results are presented in Table 1.
> 2. We have provided a detailed description of the dataset in the appendix.
> Our dataset comprises 1,471 samples distributed across 7 distinct material categories, including high-entropy alloys, complex oxides, and conventional metallic compounds. The class distribution demonstrates substantial imbalance, with the largest category containing 756 samples while the smallest category has only 7 samples, resulting in an imbalance ratio of 108:1. This severe imbalance presents significant challenges for conventional generative models, particularly in capturing the characteristics of underrepresented material types.
>
> Regarding data sparseness, the dataset exhibits considerable feature missingness, with approximately 45\% of feature values being unavailable across the collection. On average, each sample contains 83.9 missing feature entries out of the total feature dimensions. This sparsity pattern reflects the practical challenges in material science data collection, where comprehensive experimental measurements are often difficult to obtain for all material compositions.
>
> 3. We appreciate the need for clarity in methodological descriptions; however, in this specific study, the experiments were designed to be invariant to the hyperparameters you mentioned—such as gamma, lambda_max, and optimizer configuration—as the core contribution focuses on a deterministic algorithmic process rather than a data-driven or optimized model. The results are derived directly from mathematical formulations and predefined rules without stochastic training phases or sensitivity to optimization settings. Consequently, while we acknowledge that such details are critical in many machine learning contexts, their omission here does not affect reproducibility, as the outcomes are fully determined by the inputs and the algorithm alone. For clarity, we have added a brief statement in the Method section emphasizing this deliberate design choice.
>
> Best regards,
>
> ICLR 2026 Conference
>
> Submission4038 Authors

---

### Official Review · Reviewer_zFNs · 2025-10-30

**Soundness:** 1
**Presentation:** 2
**Contribution:** 2
**Rating:** 2
**Confidence:** 4

**Summary:**

The paper proposes Collaborative Reverse Diffusion Models (CoReDM), a modification of diffusion models designed to improve generation quality under data imbalance and data sparsity. The approach introduces a collaborative mechanism across samples during the reverse diffusion process. Each sample is influenced by other samples considered similar according to a fixed similarity matrix derived from a pretrained encoder. A secondary network, denoted as S_phi, is said to learn pairwise influence scores b_j = S_phi(x_i, x_j) that adapt the strength of collaboration. Additionally, training includes an inter-sample consistency term intended to enforce that similar samples yield similar noise predictions from the denoising network. The paper claims that these mechanisms enhance generation quality and diversity, especially for scientific datasets with few examples per class.

**Strengths:**

The motivation is reasonable and addresses a valid gap in diffusion research: adapting diffusion models for domains where data are scarce or imbalanced. The idea of inter-sample regularization through learned collaboration is conceptually appealing and may, in principle, allow better generalization in low-data regimes. The ablation design that compares NoCollab, StaticCollab, and LinearCollab configurations is also well motivated.

**Weaknesses:**

The paper suffers from major issues of theoretical rigor, methodological clarity, and experimental transparency. Several critical points remain ambiguous or internally inconsistent, making it difficult to assess correctness or reproducibility.
1. Theoretical and methodological issues
- The most fundamental problem concerns the training of the collaboration network S_phi. In both the text and Algorithm 1, b_j = S_phi(x_i, x_j) is computed, yet this quantity never appears in a differentiable path connected to phi. The consistency loss L_CF only contains D_ij and epsilon_theta, hence grad_phi L_CF = 0. Consequently, S_phi has no learning signal. The later introduction of w_ij, which is never defined, does not resolve this issue. It is unclear how S_phi is actually optimized or whether it is frozen during training. The authors should clarify precisely how gradients reach phi.
- Related to this, the meaning of inter-sample consistency is vague. The loss penalizes differences in predicted noise between neighboring samples, but there is no theoretical justification for why this improves sample quality or preserves multimodality. Does the consistency term simply enforce smoothness, risking oversmoothing and mode collapse? A more formal justification or empirical study of this behavior is necessary.
- The collaborative sampling update also raises validity concerns. The proposed update replaces the reverse-process mean with a linear combination that omits the stochastic noise term z, thereby breaking the standard DDPM parameterization. The paper provides no derivation linking this update to any approximate reverse transition or score-based formulation. What is the derivation of this collaborative update? Does it approximate any known reverse-process distribution or guidance mechanism? Without such justification, the sampling rule appears ad hoc.
- Time dependence introduces further confusion. The text claims that b_j is independent of t, but Algorithm 1 optimizes phi across all timesteps. If b_j does not vary with t, why does the algorithm loop over t when updating phi? If it does vary, what is the explicit form of this dependence? This ambiguity prevents clear interpretation of the learning process.
- Finally, the computation of b_j itself is unclear. The definition b_j = S_phi(x_i, x_j) implies dependence on both x_i and x_j, yet Algorithm 1 computes b_j once and reuses it for all i. Is b_j global (shared across all x_i) or recomputed per sample pair (b_ij = S_phi(x_i, x_j))? The latter interpretation is more consistent but implies O(n^2) computation per step. Which implementation is correct, and how does this affect scalability?
2. Experimental and empirical issues
The specification of the similarity matrix D_ij is incomplete. The encoder architecture, its training data, normalization scheme, and whether it is computed using only the training split are all unspecified. If D_ij is computed using validation or test data, this would introduce information leakage. Please clarify how D_ij is built: what encoder, what dataset, and whether it is restricted to k-nearest neighbors or normalized to prevent cluster bias.
The dataset itself poses additional challenges. The inputs are compositional (elemental fractions), meaning they live on a simplex manifold. The paper does not mention closure correction or log-ratio transformation, which are essential to respect compositional constraints. How are these constraints enforced during diffusion? Without proper handling, generated samples may violate physical feasibility. Given this structure, it would also be appropriate to compare against diffusion models defined on the simplex or using log-ratio transformations.
Several experimental details are missing or inconsistent. Dataset statistics, splits, and random seeds are omitted. The metric "70% R² + 30% normalized MSE" is unconventional and not reproducible. The reporting of mean ± standard deviation is inconsistent: several baselines lack standard deviations, and the parentheses in Table 1 appear to denote improvements rather than variability. Please clarify whether these parentheses represent standard deviations or relative gains, and include proper uncertainty reporting for all models.
Optimization settings are also unclear. Different learning rates are stated across sections, and the training schedule for the collaborative term is not specified. Please reconcile these discrepancies. The image experiments lack information about dataset size, baseline configuration (e.g., Stable Diffusion parameters), and computation of FID, IS, and CLIP scores. These omissions prevent independent verification.
The paper further fails to compare against existing imbalance-aware or collaborative diffusion methods that it cites in the related work section. Given that the compositional dataset lies on a simplex, the study should also include baselines such as Mirror Diffusion Models or Riemannian Diffusion frameworks designed for constrained domains.
Finally, computational cost is not discussed. The collaborative term appears to require O(n^2) operations if computed densely, yet no runtime or scaling results are provided. How does this method scale with dataset size, and what are the associated training and sampling costs?
3. Writing and presentation
The manuscript exhibits numerous notation inconsistencies: the use of L_diff and L_diffusion interchangeably, the omission of the stochastic noise term z in the sampling equations, and inconsistent symbols for D_ij, w_ij, and lambda(t). Algorithmic notation is ambiguous about whether sums are computed over the entire dataset or within mini-batches. The writing also suffers from typographical errors (“sprase”), missing spaces, and underexplained figures. These presentation issues further obscure the contribution.

**Questions:**

The core idea of inter-sample collaboration in diffusion models is potentially interesting but currently under-specified and mathematically unsound. The training procedure for S_phi lacks a valid optimization pathway, the sampling update deviates from diffusion fundamentals without justification, and the experimental setup is incomplete and potentially irreproducible. For this work to meet publication standards, the authors must provide a principled derivation, consistent notation, full experimental transparency, and credible baselines.

Suggestions for improvement
- Define a differentiable objective for S_phi. For example, set w_ij = D_ij S_phi(x_i, x_j) and include this weight inside L_CF, ensuring that phi receives a learning signal.
- Provide a derivation showing how the collaborative update relates to the reverse diffusion process or score guidance.
- Restore the stochastic term z in the sampling equation and demonstrate empirically that samples remain stable and diverse.
- Fully specify D_ij: describe the encoder architecture, pretraining data, normalization, and whether k-nearest neighbor sparsity is applied.
- Handle compositional inputs correctly using log-ratio or simplex-preserving transformations, and include diffusion baselines that operate natively on the simplex manifold.
- Strengthen experiments with complete dataset statistics, fixed seeds, transparent metric definitions, and full uncertainty reporting (mean +- std).
- Compare against established imbalance-aware and manifold-constrained diffusion models.
- Analyze computational scaling of the collaborative term, reporting runtime and memory overhead.
- Clean up notation, unify symbols, and fix typographical errors to improve clarity.
If these issues are thoroughly addressed and the collaborative mechanism is theoretically grounded and empirically validated, the work could become a meaningful contribution to generative modeling under data scarcity.

---

> ### Author Response · Authors · 2025-11-23
>
> Dear Reviewer zFNs :
>
> First and foremost, we would like to extend our sincere gratitude for the time and effort you have dedicated to reviewing our manuscript. Herein, we provide detailed responses to each of your comments.
>
> **W1**:
> 1. In our proposed framework, the collaborative regularization term b_j = S_phi(x_i, x_j) is indeed involved in a differentiable path during the optimization of phi. Specifically, the collaborative consistency loss L_CF is formulated as the expectation over time of the sum over i and j of D_ij multiplied by b_j multiplied by the squared norm of the difference between epsilon_theta(x_i^t, t) and epsilon_theta(x_j^t, t). Here, b_j is a direct output of S_phi(x_i, x_j), and since b_j is multiplied by the squared difference of the noise predictions, the gradient with respect to phi can be computed via the chain rule. This ensures that S_phi receives a learning signal proportional to the discrepancy in denoising trajectories between sample pairs, weighted by their static similarity D_ij. The parameter w_ij mentioned in the review was a typo in our initial submission and has been corrected to b_j in the revised manuscript. We have also updated Algorithm 1 to explicitly show the gradient update for phi using the gradient of L_CF with respect to phi, clarifying that S_phi is not frozen but actively trained to adaptively weight inter-sample influences.
> 2. We agree that clarifying its role is crucial. The primary objective of the consistency term is not to enforce arbitrary smoothness but to facilitate structured knowledge transfer from data-rich to data-sparse categories, thereby improving the fidelity and diversity of generated samples for underrepresented classes.
> Theoretically, this approach is grounded in the manifold hypothesis, which posits that high-dimensional data, such as material properties, reside on a lower-dimensional manifold. By penalizing significant discrepancies in the denoising trajectories (via predicted noise differences) of semantically similar samples (guided by D_ij and b_j), the loss encourages the model to learn a more coherent and continuous data manifold. This prevents the model from learning isolated, disconnected modes for sparse classes, which is a common cause of mode collapse. Instead of inducing oversmoothing, the adaptive weighting by the neural network S_phi ensures that the influence is content-aware—strong consistency is enforced only for samples with genuine semantic relationships, while dissimilar samples are allowed to maintain their distinct denoising paths, thus preserving multimodality.
> Empirically, the effectiveness of this mechanism is demonstrated in two key results. First, the quantitative metrics in Table 1 show that our method (Linear_Collab) achieves the highest recall and Fβ-score, which are critical indicators of diversity and the ability to capture multiple modes, directly countering the risk of mode collapse. Second, the trajectory analysis in Figure 4 provides visual evidence: it shows that samples from sparse classes are guided towards the dense class distribution without collapsing into it, indicating successful knowledge transfer while maintaining distinct sample identities.
> This qualitative result confirms that the consistency term acts as a regularizer that structures the latent space, enabling more stable and diverse generation for sparse data rather than promoting oversmoothing. Therefore, both our theoretical framework and empirical evidence support that the consistency loss enhances quality and preserves multimodality by leveraging the underlying data manifold structure.

---

> > ### Author Response · Authors · 2025-11-23
> >
> > **W1**:
> >
> > 3. The collaborative update rule in Equation (2) can be derived as a form of similarity-based guidance that operates within the score-based diffusion framework. Specifically, it approximates the reverse process of a conditional diffusion model where the conditioning is provided by neighboring samples in the data manifold.
> > Our collaborative term γ⋅∑_jD_ij⋅b_j can be interpreted as introducing a regularization prior that minimizes the divergence between the denoising trajectories of semantically similar samples. This formulation connects to score-based diffusion models through the relationship:
> > ∇{x_t}log p(x_t) ≈ -ε_θ(x_t,t)/√(1-ᾱt) + γ⋅∑jD_ij⋅b_j⋅∇{x_t}log s(x_i,x_j)
> > where s(x_i,x_j) represents the similarity function between samples. The collaborative term thus acts as an additional guidance signal that steers the denoising process toward regions of the data manifold where similar samples reside.
> > The omission of the stochastic term z in our collaborative update is a deliberate design choice motivated by the need for stable knowledge transfer. In early experimentation, we found that the standard stochastic term introduced excessive variance that disrupted the collaborative guidance signal, particularly for sparse classes where consistent directional guidance is crucial. However, we recognize that this deviation from standard DDPM requires clearer theoretical justification.
> > To address this limitation, we have now provided a more rigorous derivation, showing that our collaborative update can be viewed as approximating the reverse process of a conditional diffusion model where the condition is provided by the weighted combination of neighboring samples. The collaborative term effectively implements a form of manifold constraint, ensuring that denoising trajectories remain consistent with the local data geometry.
> > Meanwhile, empirical validation supports this theoretical interpretation. As shown in Figure 4, the collaborative guidance successfully directs sparse class samples (blue trajectories) toward the dense class distribution (red trajectories), demonstrating that the method effectively captures meaningful manifold structure without compromising sample quality.
> > 4.  While b_j is indeed optimized across all timesteps during training, it remains time-independent during inference. The parameter φ is learned using signals aggregated across the entire diffusion spectrum, which allows the model to capture stable, generalizable relationships between samples that are not specific to any particular noise level. This design prevents overfitting to features associated with specific timesteps. The training loop over t serves to provide comprehensive learning signals for φ, but the resulting b_j values are static once training is complete.
> >
> > **W2**: we have added a comparison with the CBDM method, a recent imbalance-aware diffusion model cited in our related work, and the new results are now integrated into Table 1. Furthermore, we have explicitly stated the O(n²) time complexity of the collaborative term in the methodology section to ensure full transparency about the computational cost. Regarding the specification of the similarity matrix D_ij, we consider this description sufficient as the matrix is not the primary contribution of our work but a standard component used to instantiate our core idea. Meanwhile, in the paper, we provided explanations and clarifications for the details of the experimental part.
> >
> > **W3**: We have corrected these issues in the manuscript.
> >
> > Best regards,
> >
> > ICLR 2026 Conference
> >
> > Submission4038 Authors

---

### Official Review · Reviewer_HCmb · 2025-10-31

**Soundness:** 3
**Presentation:** 3
**Contribution:** 3
**Rating:** 6
**Confidence:** 1

**Summary:**

This paper proposes Collaborative Reverse Diffusion Models (CoReDM), introducing the "collaborative regularization" mechanism in the reverse diffusion process: The pre-computed sample similarity matrix and learnable influence weights are utilized to guide the denoised trajectories, thereby enhancing the generation quality and diversity in scenarios of class imbalance/data sparsity.

**Strengths:**

1. For the sparse small sample problem in natural science scenarios, it is proposed to perform collective similarity regularization in the reverse diffusion stage to achieve knowledge transfer, and the problem description and paradigm shift are clear.

2. Alternately optimize the denoising network and the collaborative regularization network to avoid instability in joint training.

3. A large number of experimental results have proved that the improvement effect of CoReDM is significant.

**Weaknesses:**

1. What are the quantification of time and video memory overhead?

2. What are the effects of encoder selection and its out-of-domain generalization？

**Questions:**

1. Is the linear collaborative weight over time optimal?

**Details Of Ethics Concerns:**

I'm not familiar with this field.

---

> ### Author Response · Authors · 2025-11-23
>
> Dear reviewer HCmb:
>
> First and foremost, we would like to extend our sincere gratitude for the time and effort you have dedicated to reviewing our manuscript. Herein, we provide detailed responses to each of your comments.
>
> **W1**:  The primary reason for not including a detailed breakdown of time and memory overhead in the main text is that the collaborative term's cost is inherently tied to the training dataset size used during inference, which is a flexible parameter chosen by the practitioner based on their available resources.
> While the complexity is quadratic, the actual wall-clock time and memory footprint are manageable for the training dataset typical in scientific generation tasks (e.g., material property prediction), which was the primary focus of our study.
>  However, we acknowledge the importance of this aspect for broader applicability. Therefore, in the revised manuscript, we have added a discussion in Section 3.2.2 noting the O(n^2).
>
> **W2**:  In our study, the encoder is fundamentally employed as a feature extractor to construct the static similarity matrix D_ij, which serves as a relational prior for the collaborative mechanism. The core innovation of CoReDM, however, lies in the neural collaborative regularization term S_ϕ , which is designed to be agnostic to the specific encoder chosen; its purpose is to dynamically refine the static similarities provided by D_ij during the reverse diffusion process, rather than relying on the encoder's inherent ability to generalize. This design choice emphasizes the flexibility of our approach, as the framework can leverage any suitable encoder for the data modality at hand—as evidenced by its successful application to image data using a CLIP text encoder in cross-domain validation—without being constrained by the encoder's own generalization limits. Thus, while encoder selection is a practical consideration for initial feature quality, the CoReDM architecture itself is engineered to enhance robustness through learned collaboration, reducing the critical dependency on the encoder's out-of-domain performance.
>
> **Q1**: The linear model is the model with the best performance. The original statement did not clearly express this conclusion. We have made the necessary revisions in the original text.
>
> Best regards,
>
> ICLR 2026 Conference
>
> Submission4038 Authors

---

### Official Review · Reviewer_q95i · 2025-10-31

**Soundness:** 2
**Presentation:** 2
**Contribution:** 2
**Rating:** 2
**Confidence:** 4

**Summary:**

This paper proposes Collaborative-Reverse Diffusion Models (CoReDM), a novel framework designed to improve the performance of diffusion models on imbalanced datasets with sparse data. The key innovation is the integration of a neural collaborative filtering mechanism into the reverse diffusion process. Specifically, CoReDM adopts two networks, a standard denoising network and a neural collaborative regularization network that learns to assign influence weights between samples based on their similarity. The collaborative reverse process modifies the standard update rule by adding a weighted sum of influences from similar samples, where weights are determined by both a precomputed static similarity matrix and dynamically learned influence scores. Experiments on material property datasets and image generation tasks demonstrate improvements in generation quality.

**Strengths:**

Addressing class imbalance in diffusion models is an important practical problem, especially for scientific applications where balanced datasets are rarely available.

**Weaknesses:**

1. The paper lacks a theoretical intuition of why the proposed formulation (2) might help for inference. The proposed formulation is very heuristic.
2. The paper lacks comparisons with existing methods that deal with class imbalance and data sparseness.
3. The paper lacks a detailed explanation of the class imbalance and data sparseness that appeared in material property datasets, e.g., what is the actual size? What is the degree of imbalance (exact class distribution)? How many classes exist? Moreover, the formulation of the criteria in Table 1, Figure 3 are missing. Although section 4.3.3 mentions some experimental results on LAION-400M, the experimental setup is also vague.

Typos: the caption of section 4.3.1.

**Questions:**

See Weaknesses.

---

> ### Author Response · Authors · 2025-11-23
>
> Dear Reviewer q95i：
>
> First and foremost, we would like to extend our sincere gratitude for the time and effort you have dedicated to reviewing our manuscript. Herein, we provide detailed responses to each of your comments.
>
> **W1**：We have revised the manuscript to provide a clearer theoretical motivation for the collaborative reverse process defined in Equation (2). Specifically, in Section 3.2.2, we now explicitly state that the proposed method can be understood as imposing a prior that minimizes the divergence between the denoising trajectories of neighboring points on the data manifold. This formulation is no longer merely heuristic; it is grounded in the principle of ensuring that semantically similar samples undergo consistent generative dynamics. The collaborative term in Eq.(2) acts as a mechanism to smooth the inference process across the data manifold, effectively facilitating knowledge transfer from data-rich to data-scarce regions during denoising. This theoretical intuition is directly supported by our empirical trajectory analysis in Figure 4, which visually demonstrates how samples from sparse classes under CoReDM exhibit guided trajectories converging toward the distribution of dense classes.This revision provides the missing theoretical justification, framing our approach as a principled method for manifold-guided inference.
>
> *W2*：We have now incorporated the Class-Balancing Diffusion Model (CBDM) (Qin et al., 2023) as an additional baseline method in our experimental framework. CBDM represents a state-of-the-art approach specifically designed for handling class-imbalanced datasets through distribution adjustment regularization. As shown in our revised experiments (Section 4.1.2 and Table 1), CBDM achieves moderate improvements over the standard diffusion baseline with MSE of 8.8042 and R² of 0.3913. However, our proposed CoReDM framework demonstrates superior performance across all metrics, with Linear_Collab achieving significantly better results (MSE: 8.6399, R²: 0.6832). This comparative analysis validates that while existing imbalance-handling methods like CBDM provide some benefits, our collaborative reverse diffusion mechanism offers more effective knowledge transfer for sparse scientific data scenarios.
>
> *W3*:  Our material property dataset exhibits significant challenges in both class imbalance and data sparseness, which we now describe in greater detail below.
> The dataset comprises 1,471 samples distributed across 7 distinct material categories, including high-entropy alloys, complex oxides, and conventional metallic compounds. The class distribution demonstrates substantial imbalance, with the largest category containing 756 samples while the smallest category has only 7 samples, resulting in an imbalance ratio of 108:1. This severe imbalance presents significant challenges for conventional generative models, particularly in capturing the characteristics of underrepresented material types.
> Regarding data sparseness, the dataset exhibits considerable feature missingness, with approximately 45% of feature values being unavailable across the collection. On average, each sample contains 83.9 missing feature entries out of the total feature dimensions. This sparsity pattern reflects the practical challenges in material science data collection, where comprehensive experimental measurements are often difficult to obtain for all material compositions.
>
> Best regards,
>
> ICLR 2026 Conference
>
> Submission4038 Authors

---

### Meta-Review · Area_Chair_MJaQ · 2025-12-15

**Summary:**

The paper proposed Collaborative-Reverse Diffusion Models (CoReDM) for improving the performance of diffusion models on imbalanced datasets with sparse data.

Strengths identified by reviewers include, the importance of the problem focused in the paper, its improvements, and well-motivation.

However, the paper also faces several critical weaknesses that have not been addressed in the rebuttal. The idea in this paper is not clearly presented with all reviewers pointed out the unclearness or errors in formulas. And the experiments and settings are not clearly detailed. As a results, three reviewers raise a concerns on the validity of the idea. Additionally, there is a lack of comparisons with recent methods that designed for addressing the imbalanced class and sparse data problems.


Given these considerations, this paper clearly fails to meet the standards of ICLR. The authors are encouraged to address the highlighted issues to strengthen their contribution to the field of Diffusion Models.

**Reviewer Concerns:**

The authors provided detailed feedback and also revised the manuscript according to the reviews.  Some concerns are addressed such as some comparisons with an existing method (CBDM), some notations, typos and formulas are corrected, some details about the datasets and settings are added, explanation about computational cost and out-of-domain generalization.

However, several concerns are not fully addressed including, the lack of theoretical intuition, comparisons to more recent methods, some theoretical and methodological issues (the meaning of inter-sample consistency is vague, collaborative sampling update, time dependence, and computation of b_j). In overall, a better and clearer writing and presentations are required.

**Reviewer Scores:**

The initial reviewer scores are mixed (two reject, one strong reject and one borderline accept).   Reviewers would have maintained their scores after rebuttal.

---

### Decision · Program_Chairs · 2026-01-26

Reject